# NEGATIVE SAMPLING IN VARIATIONAL AUTOENCODERS

## ABSTRACT

We propose negative sampling as an approach to improve the notoriously bad out-of-distribution likelihood estimates of Variational Autoencoder models. Our model pushes latent images of negative samples away from the prior. When the source of negative samples is an auxiliary dataset, such a model can vastly improve on baselines when evaluated on OOD detection tasks. Perhaps more surprisingly, we present a fully unsupervised version of employing negative sampling in VAEs: when the generator is trained in an adversarial manner, using the generator's own outputs as negative samples can also significantly improve the robustness of OOD likelihood estimates.

## 1 INTRODUCTION

Learning semantically meaningful and useful representations for downstream tasks in an unsupervised manner is a big promise of generative modeling. While a plethora of work demonstrates the effectiveness of deep generative models in this regard, recent work of Nalisnick et al. (2019a) and Choi et al. (2018) show that these models often fail even at a task that is supposed to be close to their original goal of learning densities. Variational Autoencoders, PixelCNN and flow-based models cannot distinguish common objects like cats and dogs from house numbers. That is, when trained e.g., on CIFAR-10, the models consistently assign higher likelihoods to the elements of the SVHN test set than for the elements of the CIFAR-10 test set or even the elements of the CIFAR-10 train set. As generative models are becoming more and more ubiquitous due to the massive progress in this area in recent years, it is of fundamental importance to understand these phenomena.

In this work we study Variational Autoencoder (VAE) models, and besides the likelihood, we also investigate to what extent the latent representation of a data point can be used to identify out-of-distribution (OOD) samples (points that are not from the true data distribution). For this purpose, we consider the KL divergence between the prior and the posterior distribution of a data point as a score to distinguish inliers and outliers. Our contributions are summarized as follows:

- We demonstrate empirically that the extent of this notorious phenomenon — of bad out-of-distribution likelihood estimates — present in VAEs largely depends on the observation model of the VAE. In particular, our experiments show that it diminishes when a Gaussian noise model is considered (with a reasonably sized fixed or learned variance) instead of a Bernoulli. Meanwhile, when examining only the KL divergence between the prior and the posterior distributions in the latent space (instead of the full likelihood), the weak separating capability more consistently prevails between inliers and outliers.

- We propose *negative sampling in Variational Autoencoders* as an approach to alleviate the above weaknesses of the model family. In this method, we introduce an additional prior distribution $\bar{p}(\boldsymbol{z})$ in the latent space, where the representations of negative samples are meant to be mapped by the inference model of the VAE machinery. Negative samples can be obtained from an auxiliary dataset, or — to remain completely in the unsupervised setting — from a generative model trained on the ground truth distribution itself.

- We present empirical evidence that utilizing negative samples either from an auxiliary dataset or from an adversarial training scheme (using the adversarially trained generative model itself to provide the negative samples) significantly and consistently improves the discriminative power of VAE models regarding out-of-distribution samples.

The general intuition behind our approach is that if the posterior distribution of each and every point is pulled towards the prior then it is rather natural to expect that the system will map out-of-distribution samples close to the prior, as well. This viewpoint suggests that providing negative signals throughout the learning process would be beneficial to enhance the OOD discriminative power of the system.

Hendrycks et al. (2019) demonstrate that utilizing auxiliary datasets as OOD examples (as a supervised signal) significantly improves the performance of existing anomaly detection models on image and text data. First, we study how this approach can be employed in the VAE setting. Beyond that, we also propose a method which remains completely in the unsupervised learning paradigm (without using an auxiliary dataset for supervised signal). The core idea of this unsupervised approach is to use a generative model to provide near-manifold negative samples throughout the training process for which the model is either implicitly or explicitly encouraged to give low likelihood estimates. In our proposed method, these negative samples are obtained from the currently trained VAE model itself by utilizing the generated samples.

## 2 BACKGROUND

The generative modeling task aims to model a ground truth data density $p^*(\boldsymbol{x})$ on a space $\mathcal{X}$ by learning to generate samples from the corresponding distribution. The learning is done in an unsupervised manner with sampled observables $\mathbf{X} = \{\mathbf{x}^{(i)}\}_{i=1}^N$ as training points assumed to be drawn independently from $p^*(\boldsymbol{x})$, where $N$ is the sample size. In latent variable models, the observables are modeled together with hidden variables $\boldsymbol{z}$ on which a prior distribution $p(\boldsymbol{z})$ is imposed.

The Variational Autoencoder (VAE) (Kingma & Welling, 2013) is a latent variable model that takes the maximum likelihood approach and maximizes a lower bound of the sample data log likelihood $\sum_{i=1}^N \log p_\theta(\mathbf{x}^{(i)})$, where $\theta$ are the generator parameters. The utilized lower bound $\mathcal{L}(\theta, \phi, \mathbf{x}^{(i)})$ (called the ELBO) comes from a variational approximation $q_\phi(\boldsymbol{z}|\mathbf{x}^{(i)})$ of the intractable posterior $p_\theta(\boldsymbol{z}|\mathbf{x}^{(i)})$, where $\phi$ are the variational parameters:

$$\log p_\theta(\mathbf{x}^{(i)}) = \log \int p_\theta(\mathbf{x}^{(i)}|\boldsymbol{z})p(\boldsymbol{z}) \geq$$
$$\geq \underbrace{\mathbb{E}_{q_\phi(\boldsymbol{z}|\mathbf{x}^{(i)})} \log p_\theta(\mathbf{x}^{(i)}|\boldsymbol{z})}_{\text{Reconstruction term}} - \underbrace{D_{\mathrm{KL}}(q_\phi(\boldsymbol{z}|\mathbf{x}^{(i)}) \parallel p(\boldsymbol{z}))}_{\text{KL divergence term}} \triangleq \mathcal{L}(\theta, \phi, \mathbf{x}^{(i)}).$$

In the VAE model the parametrized distributions $p_\theta$ and $q_\phi$ are modeled with neural networks and are trained jointly to maximize $\mathcal{L}$ with some variant of the SGD. The prior is often chosen to be the multivariate standard normal distribution, and a Bernoulli or Gaussian noise model is considered in the observable space to define the likelihood.

To give likelihood estimates for unseen data points at test time, one can use the trained inference model $q_\phi(\boldsymbol{z}|\mathbf{x}^{(i)})$ (also referred to as encoder) and generative model $p_\theta(\mathbf{x}^{(i)}|\boldsymbol{z})$ (also referred to as decoder) to estimate the ELBO, thus giving a lower bound of the likelihood. Throughout our paper, we are considering these ELBO estimates to measure the likelihood of data points.

## 3 NEGATIVE SAMPLING IN VARIATIONAL AUTOENCODERS

To incorporate negative samples in the VAE training process, we introduce an additional prior distribution $\bar{p}(\boldsymbol{z})$ (called the *negative prior*) on the latent variables $\boldsymbol{z}$ into which the representations of *negative samples* $\overline{\mathbf{X}} = \{\overline{\mathbf{x}}^{(i)}\}_{i=1}^M$ are meant to be mapped by the inference model. This is encouraged in the training process by adding to the regular ELBO a new loss term: the KL divergence of the posterior distributions of negative samples to this negative prior. Thus the joint loss function (to be minimized) is as follows:

$$L(\theta, \phi, \mathbf{x}^{(i)}, \bar{\mathbf{x}}^{(i)}) \triangleq -\mathcal{L}(\theta, \phi, \mathbf{x}^{(i)}) + D_{\mathrm{KL}}(q_\phi(\mathbf{z}|\bar{\mathbf{x}}^{(i)}) \parallel \bar{p}(\mathbf{z})) =$$
$$= \underbrace{-\mathbb{E}_{q_\phi(\mathbf{z}|\mathbf{x}^{(i)})} \log p_\theta(\mathbf{x}^{(i)}|\mathbf{z}) + D_{\mathrm{KL}}(q_\phi(\mathbf{z}|\mathbf{x}^{(i)}) \parallel p(\mathbf{z}))}_{-1 \cdot \mathrm{ELBO\ for\ } \mathbf{x}^{(i)}} + \underbrace{D_{\mathrm{KL}}(q_\phi(\mathbf{z}|\bar{\mathbf{x}}^{(i)}) \parallel \bar{p}(\mathbf{z}))}_{\mathrm{KL\ term\ for\ negative\ sample\ } \bar{\mathbf{x}}^{(i)}}. \quad (1)$$

**Motivating our loss function**   The loss function defined in equation 1 is still an upper bound of the negative data log likelihood (for the positive samples) as the added loss term is non-negative. The new loss term explicitly imposes the discriminative task for the inference model: to distinguish inliers and outliers in the latent space. With these two components, while still preserving the aim of maximizing the likelihood for inliers, we also expect the implicit behavior of reducing likelihood estimates for outliers. For the outliers, a trained inference model produces latent representations that are close to the negative prior $\bar{p}(\mathbf{z})$, thus, supposedly far from the prior $p(\mathbf{z})$. Also, the system is not encouraged to learn to generate from the vicinity of the negative prior, therefore not only the KL term of the likelihood, but the reconstruction part of a negative sample is affected when inferring the likelihood estimate of an outlier.

### 3.1   THE CHOICE OF THE NEGATIVE PRIOR

One has numerous options to choose the positive and negative priors. In this paper, we simply choose to use a standard normal for the positive prior, and a shifted standard normal for the negative prior. With a rotationally symmetric posterior distribution, the distance between the two priors would be the only unspecified hyperparameter of such a model. The assumption of diagonal covariance matrix posterior breaks rotational symmetry in principle, but our exploratory experiments have demonstrated that the magnitude of the shift is a more significant modeling choice than the direction/sparsity of the shift.

**The role of** $D_{\mathrm{KL}}(\bar{p}(\mathbf{z}) \parallel p(\mathbf{z}))$   The magnitude of KL divergence between the negative and the positive prior plays an important role. Larger $D_{\mathrm{KL}}(\bar{p}(\mathbf{z}) \parallel p(\mathbf{z}))$ values result in larger $D_{\mathrm{KL}}(q_\phi(\mathbf{z}|\bar{\mathbf{x}}^{(i)}) \parallel p(\mathbf{z}))$ terms when evaluating the KL divergence term of the likelihood in a trained model, and also result in heavier weighted KL divergence terms during the optimization process. E.g., with a farther shifted negative prior mean, a larger penalty is given for a wrong inference.

**The role of the latent dimension**   The above argument gives rise to an interesting side effect: increasing the latent dimension also increases $D_{\mathrm{KL}}(\bar{p}(\mathbf{z}) \parallel p(\mathbf{z}))$, thus resulting in a larger weight of the discriminative KL terms.

With the above simple choice of the shifted normal for the negative prior, our experiments already demonstrate the effectiveness of our proposed method. One possible direction for further improvement would be to explore positive-negative prior pairs that better reflect the inlier-outlier structure of different datasets. We leave this investigation for future work.

### 3.2   SOURCE OF NEGATIVE SAMPLES

Negative samples can also be obtained in different ways. The task of our models is to generalize from the negative samples as much as possible to all possible out-of-distribution samples, so that they can push down likelihood estimates of those. Depending on the source of negative samples, this generalization can be easier or harder. We conduct experiments with several variants:

- samples from an auxiliary dataset,
- the data with isotropic Gaussian noise added,
- generated samples from the trained model itself,
- generated samples utilizing an adversarial training scheme.

Negative samples that are very far from the data manifold do not facilitate generalization. Noise added to data points is a simple and principled way to sample from the vicinity of the data manifold, but as we will see, it does not provide good generalization. We argue that the reason for this is that

discriminating between noisy and noiseless points is too easy for the encoder, so "semantically" the noisy versions are far from the data manifold. In contrast, utilizing samples produced by a generative model (which could be the trained generative model itself) is a more suitable way to acquire near-manifold negative samples, as we will experimentally demonstrate.

**Why using generated data as negative samples could help?** An immediate counterargument against utilizing generated samples as negatives could be the following: for a well-trained model, the generated images are indistinguishable from the ground truth images, so training the model to discriminate them is nonsensical. There are several reasons why such an unsupervised method could still work. First, in practice, a trained generative model is typically not perfect. True data samples and generated samples can be distinguished even for fully trained models. Second, even assuming a perfect generator at the end, during the training process, the generated samples might still help to guide the model toward an equilibrium that promotes a lower likelihood for OOD samples. Moreover, when utilizing auxiliary datasets, we have to choose the auxiliary dataset (or multiple datasets) carefully to wedge in between the training set and a potential out-of-distribution data point, otherwise the weak separating capability could prevail. In contrast, learning to discriminate the generated near-manifold examples from the ground truth data is a harder task, and could result in discriminating a more diverse set of potential out-of-distribution samples.

Our experiments also confirm this. When utilizing an adversarial training scheme (to be introduced later in this section), generated images not only facilitate our discriminative training procedure, but even achieve a higher level of generalization in the following sense: utilizing generated images improves on the baseline in *all permutations* of the roles for the grayscale datasets when considering discrimination in the latent space, while utilizing auxiliary datasets fails to achieve notable improvement in some cases. (See the results in Table 1 in rows with AUC KL, and more details in the experiments section.)

In our preliminary experiments, we observed that in some examined cases, utilizing simply the generated images of the currently trained VAE model fail to provide a good signal for the discriminative task. We achieved greater success when we augmented our model with adversarial training. We hypothesize that the reason behind this is that obtained negative samples are richer in features and semantically more meaningful for the task. (See Lee et al. (2018) for an incarnation of this idea in the context of classification and generative adversarial networks.)

**The utilized adversarial training scheme** When experimenting with generated samples as negatives, we utilize an adversarial training scheme where the generator (and only the generator) gets an additional gradient signal through the encoder to map the randomly generated images into the prior. This is encouraged via the following additional loss term:

$$D_{\mathrm{KL}}(q_\phi(\boldsymbol{z}|\hat{\mathbf{x}}^{(i)}) \parallel p(\boldsymbol{z})),$$

where $\hat{\mathbf{x}}^{(i)}$ denotes a generated image obtained from the generator $p_\theta(\hat{\mathbf{x}}^{(i)}|\boldsymbol{z})$, where $\boldsymbol{z}$ is sampled from the prior $p(\boldsymbol{z})$. Together with the fact that the encoder also gets the generated images as negative samples, this results in an adversarial training procedure. In this setup, the separate loss functions of the encoder and generator are:

$$L_{enc}^{adv}(\theta, \phi, \mathbf{x}^{(i)}, \hat{\mathbf{x}}^{(i)}) = L(\theta, \phi, \mathbf{x}^{(i)}, \hat{\mathbf{x}}^{(i)}),$$

$$L_{gen}^{adv}(\theta, \phi, \mathbf{x}^{(i)}, \hat{\mathbf{x}}^{(i)}) = L(\theta, \phi, \mathbf{x}^{(i)}, \hat{\mathbf{x}}^{(i)}) + D_{\mathrm{KL}}(q_\phi(\boldsymbol{z}|\hat{\mathbf{x}}^{(i)}) \parallel p(\boldsymbol{z})).$$

Our utilized scheme is simple yet effective. However, it is just one of the options. Another choice would be to use a separate generative model with the specific task to provide negative samples. We invite the research community to develop methods that can provide near-manifold examples that can facilitate the training of models with better OOD likelihood properties.

## 4 EXPERIMENTAL RESULTS

**The general setup** Our main concern is the discriminative power of VAE models regarding out-of-distribution samples. Following the conventions of related work, the general experimental setup in this section is as follows: we train a model on a train set of a dataset (e.g. train set of Fashion-MNIST) and then require the model to discriminate between the test set of the train dataset (e.g. test

set of Fashion-MNIST) and the test set of an out-of-distribution dataset (e.g. test set of MNIST). During the training phase, the models do not encounter examples from the OOD dataset, only at test time are they expected to able to distinguish between inliers and out-of-distribution samples.

**Quantitative assessment**   For quantitative assessment, we use the threshold independent AUC metric calculated with the bits-per-dimension score (denoted by AUC BPD) and also with the KL divergence of the posterior distribution of a data point to the prior (denoted by AUC KL). All reported numbers in this section are averages of 5 runs with standard deviations denoted in parentheses.

**Datasets and experimental details**   We conduct experiments on two sets of datasets: color images of size 32x32 (CIFAR-10, SVHN, downscaled ImageNet) and grayscale images of size 28x28 (MNIST, Fashion-MNIST, Kuzushiji-MNIST, EMNIST-Letters). For both cases, the (positive) prior is chosen to be standard normal, and the latent dimension is set to 100 for color images, and to 10 for grayscale images. For a more detailed description of the utilized datasets, models, and training methodology, see Appendix A. We present generated samples from the models in Appendix D. The samples demonstrate that the models preserve their generative capability even after adding the extra loss terms.

**The choice of the negative prior**   In our experiments, the negative prior is a standard normal with a shifted mean. For color images it is centered at $25 \cdot \mathbf{1}$, for grayscale images it is centered at $8 \cdot \mathbf{1}$. The magnitude of the shift is set based on a parameter sweep, which was evaluated using Fashion-MNIST and MNIST in the range of $\{2, 4, 6, 8, 10\}$ for grayscale images, and using CIFAR-10 and SVHN in the range of $\{5, 10, 15, 20, 25, 30\}$ for color images. After observing a clear trend, we have chosen the mode.

### 4.1 THE EFFECTIVENESS OF NEGATIVE SAMPLING

To demonstrate the effectiveness of negative sampling we present two different sets of experiments: first we incorporate negative samples from an auxiliary dataset, second we explore the use of adversarially generated negative samples.

**Almost perfect discrimination with auxiliary datasets**   The AUC scores in Table 1 show that using the auxiliary dataset as a source of negative samples in most cases proved to result in models that are capable to distinguish nearly perfectly between inliers and OOD samples. This is also the case with color images, as experimental results in Table 2 show.

**Failure modes with auxiliary datasets**   One can observe in Table 1 that — despite the above mentioned improvements — there are cases when utilizing an auxiliary dataset fails to improve on the OOD separating capability. One example for this is when the inlier set is the EMNIST-Letters, the OOD test set is MNIST, and the utilized auxliary dataset is Fashion-MNIST (the results for this setup are in the last row of Table 1). Showing skirts and boots for the model in training time does not help discriminating between letters and numbers at test time. We hypothesize, that this as an example of the case, when the auxiliary dataset (regarding its features) does not wedge in between the inlier and the outlier test set. One possible way of improvement in this regard is to utilize several auxiliary datasets to present a more diverse set of examples for possible out-of-distribution samples in terms of features and semantic content.

Of course, the most beneficial would be to train the system to distinguish between the inliers and every possible outlier data points instead of just learning to separate only one or a specific set of auxiliary dataset. This motivates our experiments utilizing generated samples as negative, with the idea that learning to separate from near-manifold examples could facilitate a better generalization in terms of OOD detection.

**Unsupervised method: improvements in all permutations in AUC KL**   In the case of the grayscale images, the last column in Table 1 shows the effectiveness of the fully unsupervised approach: regardless of whether using a Gaussian and a Bernoulli noise model[1], the trained models

---

[1]Even though the Bernoulli noise model might not be a particularly good choice for modeling grayscale or color images, here we follow the literature when considering it as a baseline.

Table 1: Comparing the out-of-distribution discriminative power of baseline VAE models and VAE models with negative sampling on grayscale images. Numbers for all permutations with the different possible roles of the three datasets (MINST, Fashion-MNIST and EMNIST-Letters) are reported. When an auxiliary dataset is utilized, the auxiliary dataset is the one out of the three that is not utilized neither as inlier nor for OOD testing purposes.

|  | Inlier | OOD | Noise model | Baseline VAE (no negative) | Negative: auxiliary | Negative: adversarial |
|---|---|---|---|---|---|---|
| AUC BPD | Fashion-MNIST | MNIST | Bernoulli | 0.46 (0.05) | 1.00 (0.00) | 0.70 (0.13) |
| | Fashion-MNIST | MNIST | Gaussian | 0.98 (0.00) | 1.00 (0.00) | 0.80 (0.04) |
| | Fashion-MNIST | Letters | Bernoulli | 0.61 (0.01) | 0.99 (0.00) | 0.78 (0.07) |
| | Fashion-MNIST | Letters | Gaussian | 0.97 (0.00) | 1.00 (0.00) | 0.85 (0.04) |
| | MNIST | Fashion-MNIST | Bernoulli | 1.00 (0.00) | 1.00 (0.00) | 1.00 (0.00) |
| | MNIST | Fashion-MNIST | Gaussian | 0.97 (0.00) | 1.00 (0.00) | 0.98 (0.01) |
| | MNIST | Letters | Bernoulli | 0.99 (0.00) | 0.99 (0.00) | 0.98 (0.00) |
| | MNIST | Letters | Gaussian | 0.78 (0.14) | 0.93 (0.08) | 0.79 (0.04) |
| | Letters | Fashion-MNIST | Bernoulli | 0.98. (0.00) | 0.98 (0.00) | 0.99 (0.00) |
| | Letters | Fashion-MNIST | Gaussian | 0.80 (0.07) | 0.76 (0.08) | 0.93 (0.04) |
| | Letters | MNIST | Bernoulli | 0.58 (0.02) | 0.58 (0.02) | 0.73 (0.07) |
| | Letters | MNIST | Gaussian | 0.67 (0.17) | 0.58 (0.20) | 0.65 (0.04) |
| AUC KL | Fashion-MNIST | MNIST | Bernoulli | 0.61 (0.09) | 1.00 (0.00) | 0.88 (0.07) |
| | Fashion-MNIST | MNIST | Gaussian | 0.26 (0.03) | 1.00 (0.00) | 0.74 (0.05) |
| | Fashion-MNIST | Letters | Bernoulli | 0.68 (0.07) | 1.00 (0.00) | 0.84 (0.04) |
| | Fashion-MNIST | Letters | Gaussian | 0.38 (0.04) | 0.99 (0.00) | 0.79 (0.05) |
| | MNIST | Fashion-MNIST | Bernoulli | 0.73 (0.14) | 1.00 (0.00) | 0.94 (0.10) |
| | MNIST | Fashion-MNIST | Gaussian | 0.71 (0.04) | 1.00 (0.00) | 0.98 (0.01) |
| | MNIST | Letters | Bernoulli | 0.64 (0.03) | 0.76 (0.03) | 0.89 (0.02) |
| | MNIST | Letters | Gaussian | 0.54 (0.07) | 0.75 (0.08) | 0.74 (0.04) |
| | Letters | Fashion-MNIST | Bernoulli | 0.66 (0.14) | 0.54 (0.09) | 0.98 (0.00) |
| | Letters | Fashion-MNIST | Gaussian | 0.54 (0.10) | 0.49 (0.23) | 0.91 (0.05) |
| | Letters | MNIST | Bernoulli | 0.37 (0.05) | 0.45 (0.03) | 0.75 (0.06) |
| | Letters | MNIST | Gaussian | 0.36 (0.08) | 0.43 (0.10) | 0.64 (0.04) |

achieve higher AUC KL scores than the baseline in *all permutations*. The method also shows better AUC BPD scores than the baseline in most of the cases where the baseline fails (i.e., baselines with below 0.6 AUC BPD scores). One can observe that when the train set is EMNIST-Letters and the OOD set is MNIST, the separation is still not achieved with this method either. The possible reason behind this is that the visual features of these two datasets are very close to each other and it is a hard task to switch the default relation between them (note that when these two datasets switch roles, the likelihood estimates are correct). Table 2 shows that in the case of color images, the unsupervised method also achieves notable discriminative performance improving on the baseline.

**Random noise and additive isotropic Gaussian noise does not help**    We also investigated how the choice of negative samples influences the performance of the trained model. We conducted further experiments with the following negative samples: 1) Kuzushiji-MNIST[2] (KMNIST) as an another auxiliary dataset, 2) random noise (in which we sample each pixel intensity from the uniform distribution on $[0, 1]$ — modeling a dataset with less structure), 3) with an additive isotropic Gaussian noise added to the inlier dataset.

The results in Table 3 show that utilizing either KMNIST or MNIST-Letters results in perfect separation of the inliers (Fashion-MNIST) and outliers (MNIST). The weak results with random noise as negative samples show the significance of the choice of negative samples. We also experimented with utilizing the training set itself with an additive isotropic Gaussian noise as negative samples — a rather natural choice to provide near-manifold examples. With an additive noise of $\sigma = 0.25$,

---

[2]EMNIST-Letters, Kuzushiji-MNIST and Fashion-MNIST are datasets that can be utilized as drop-in replacements for MNIST.

Table 2: Comparing baseline VAEs and VAEs with negative sampling with Bernoulli, Gaussian, and Quantized Gaussian (Q. Gaussian) noise models on color image datasets.

|  | Inlier | OOD | Noise model | Baseline VAE (no negative) | Negative: auxiliary | Negative: adversarial |
|---|---|---|---|---|---|---|
| **AUC BPD** | CIFAR-10 | SVHN | Bernoulli | 0.59 (0.00) | 0.90 (0.05) | 0.81 (0.04) |
|  | CIFAR-10 | SVHN | Gaussian | 0.25 (0.02) | 0.93 (0.01) | 0.84 (0.03) |
|  | CIFAR-10 | SVHN | Q. Gaussian | 0.19 (0.00) | 0.92 (0.03) | 0.82 (0.03) |
|  | SVHN | CIFAR-10 | Bernoulli | 0.51 (0.00) | 1.00 (0.00) | 0.70 (0.03) |
|  | SVHN | CIFAR-10 | Gaussian | 0.92 (0.00) | 1.00 (0.00) | 0.75 (0.11) |
| **AUC KL** | CIFAR-10 | SVHN | Bernoulli | 0.29 (0.00) | 0.90 (0.06) | 0.81 (0.04) |
|  | CIFAR-10 | SVHN | Gaussian | 0.25 (0.01) | 0.93 (0.01) | 0.84 (0.03) |
|  | CIFAR-10 | SVHN | Q. Gaussian | 0.28 (0.01) | 0.92 (0.03) | 0.82 (0.03) |
|  | SVHN | CIFAR-10 | Bernoulli | 0.87 (0.00) | 1.00 (0.00) | 0.70 (0.03) |
|  | SVHN | CIFAR-10 | Gaussian | 0.74 (0.01) | 1.00 (0.00) | 0.74 (0.11) |

Table 3: Comparing baseline model and negative sampling with different sources for negatives. Columns correspond to different sources for negative samples. Results for the baseline (i.e., VAE without negative sampling) are indicated again in the first column for comparison. Samples from the different data sets are also depicted in the last row to show their general visual characteristics.

| Inlier Fashion-MNIST | OOD MNIST | Auxiliary dataset as negative | | | Negative: Adversarial |
|---|---|---|---|---|---|
|  |  | Random | KMNIST | Letters |  |
| AUC BPD | 0.46 (0.05) | 0.47 (0.05) | 1.00 (0.00) | 1.00 (0.00) | 0.70 (0.13) |
| AUC KL | 0.61 (0.09) | 0.56 (0.08) | 1.00 (0.00) | 1.00 (0.00) | 0.88 (0.07) |
| Test BPD | 0.30 (0.00) | 0.30 (0.00) | 0.30 (0.00) | 0.30 (0.00) | 0.47 (0.09) |
| OOD BPD | 0.35 (0.08) | 0.32 (0.04) | 1.10 (0.09) | 1.44 (0.20) | $10^{18}$ ($10^{19}$) |

the results for the AUC BPD metric is $0.44$ $(0.01)$ and $0.70$ $(0.09)$ for the AUC KL, showing weak discriminative power.

## 4.2 THE EFFECT OF THE NOISE MODEL

Examining the results for baseline VAE models (i.e., models without negative sampling) in Table 1 and Table 2, we can observe great variability in the OOD detection performance.

**The noise model greatly influences the phenomenon** The results suggest that the intriguing phenomenon in VAEs discussed by Nalisnick et al. (2019a) and Choi et al. (2018) is highly dependent on modelling choices. In the case of grayscale images, when changing the noise model from Bernoulli to Gaussian (and otherwise remaining in the same experimental setting as Nalisnick et al. (2019a)), the issue of assigning higher likelihood estimates to OOD samples simply does not occur. However, one can observe that discrimination between inliers and OOD samples based on the KL divergence between approximate posterior and prior is hardly feasible, with below-$1/2$ AUC scores. Meanwhile, with a Bernoulli noise model (also used in Nalisnick et al. (2019a)) both the likelihood-estimates and the KL divergences fail to discriminate. The other results in the table (where models are trained on MNIST) confirm the asymmetric behaviour already described by Nalisnick et al.

(2019a), that is, switching the roles of the inlier and outlier dataset affects the presence of the phenomenon. Concerning experiments with color images, the corresponding rows of Table 2 again show the importance of modelling choices. When CIFAR-10 is the training set, the phenomenon persistently occurs with Bernoulli, Gaussian and Quantized Gaussian noise models. When SVHN is the training set, one can observe again a great variability in the AUC scores.

## 5    RELATED WORK

Our investigations are mostly inspired by and related to recent work on the evaluation of generative models on OOD data (Shafaei et al., 2018; Nalisnick et al., 2019a; Choi et al., 2018; Hendrycks et al., 2019). These works report that despite intuitive expectations, generative models — including but not limited to VAEs — consistently fail at distinguishing OOD data from the training data, yielding higher likelihood estimates on unseen OOD samples.

Nalisnick et al. (2019a) examine the phenomenon in detail, focusing on finding the cause of it by analyzing flow-based models that allow exact likelihood calculation. Choi et al. (2018) also notice the above-mentioned phenomenon, while they address the task of OOD sample detection with Generative Ensembles. They decrease the weight of the KL divergence term in the ELBO (contrarily to what is promoted by the $\beta$-VAE loss function) to encourage a higher distortion penalty during training, resulting in a better performing model. This observation also confirms the importance of the noise model and the balance between the KL and the reconstruction term.

The ominous observation is presented also by Hendrycks et al. (2019), but they concentrate on improving the OOD data detection with Outlier Exposure. Their work demonstrates that utilizing samples from an auxiliary data set as OOD examples, i.e., training models to discriminate between training and auxiliary samples, significantly improves on the performance of existing OOD detection models on image and text data. However, they do not investigate the VAE model, and their general setup always requires an auxiliary dataset. Our work also sheds light on an issue with this approach: one should choose the auxiliary datasets carefully to obtain robust OOD detection.

Within the context of uncertainty estimation, Lee et al. (2018) demonstrate that adversarially generated samples improve the confidence of classifiers in their correct predictions. They train a classifier simultaneously with a GAN and require it to have lower confidence on GAN samples. For each class distribution, they tune the classifier and GAN using samples from that OOD dataset. Their method of utilizing generated samples of GANs is closest to our approach of using generated data points as negative samples, but Lee et al. (2018) work within a classification setting.

Nalisnick et al. (2019b) propose a solution that can alleviate the issue without modifying existing generative models, but the issue they aim to address (distributional shift) is very different from the standard concerns of OOD sample detection. Their model works by using the likelihood estimates coming from likelihood-based models as inputs to detect distributional shift, as opposed to using them as raw OOD sample detectors. The model operates under the assumption that at evaluation time, samples come in batches, and thus can be the inputs of statistical tests differentiating between likelihood estimates for inlier datasets and likelihood estimates for evaluation datasets. In the limiting case where the evaluation dataset has batch-size 1, the performance of this model can be compared meaningfully with our unsupervised models.

## 6    CONCLUSIONS

In this work, we studied Variational Autoencoder models and investigated to what extent the latent representations of data points or the likelihood estimates given by the model can be used to identify out-of-distribution samples. We demonstrated empirically that the extent of the notorious phenomenon of wrong out-of-distribution likelihood estimates present in VAEs is highly dependent on the observation model. We introduced negative sampling as an approach to alleviate the above weakness of the Variational Autoencoder model family. We presented empirical evidence that utilizing negative samples either from an auxiliary dataset or from an adversarial training scheme significantly and consistently improves the discriminative power of VAE models regarding out-of-distribution samples.

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

# A    EXPERIMENTAL DETAILS

## A.1    DATASETS AND PREPROCESSING

We conduct experiments with two types of data set: color images of size 32x32 and grayscale images of size 28x28. The utilized datasets are listed below.

**Datasets of grayscale images of size 28x28:**

- MNIST (LeCun et al., 2010): 28x28x1, 60.000 train + 10.000 test, 10 classes
- Fashion-MNIST (Xiao et al., 2017): 28x28x1, 60.000 train + 10.000 test, 10 classes
- Kuzushiji-MNIST (Clanuwat et al., 2018): 28x28x1, 60.000 train + 10.000 test, 10 classes
- EMNIST-Letters (Cohen et al., 2017): 28x28x1, 60.000 train + 10.000 test, 10 classes

**Datasets of color images of size 32x32:**

- CIFAR-10 (Krizhevsky, 2009): 32x32x3 images, 50.000 train + 10.000 test, 10 classes
- SVHN (cropped) (Netzer et al., 2011): 32x32x3 images, 73.257 train + 26,032 test (+ 531.131 extra unlabeled), 10 classes
- Downsampled ImageNet (van den Oord et al., 2016): 32x32x3 images, 1.281.149 train + 49.999 validation, 1000 classes

We apply no preprocessing step other than normalizing the input images to $[0, 1]$.

## A.2    NETWORK ARCHITECTURE AND TRAINING DETAILS

**Details for grayscale images**    Following Nalisnick et al. (2019a), for grayscale images, we use the encoder architecture described in Rosca et al. (2018) in appendix K table 4. Also, as in Rosca et al. (2018), all of the models are trained with the RMSProp optimizer with learning rate set to $10^{-4}$. We train the models for 100 epochs with mini-batch size of $50$. We update the parameters of the encoder and decoder network in an alternating fashion.

**Details for color images**    For color images we use a DCGAN-style CNN architecture with Conv–BatchNorm–ReLU modules for both the encoder and the decoder. The size of the kernels are $4 \times 4$, and the number of filters are $32, 64, 128$ for the encoder; and $128, 64, 1$ for the decoder. All of the models are trained with the Adam optimizer ($\beta_1 = 0.9, \beta_2 = 0.999$) for 100 epochs with mini-batch size 50. The learning rate is set to $10^{-4}$. We update the parameters of the encoder and decoder network in an alternating fashion. When generated images are used as negative samples, we employ spectral normalization (Miyato et al., 2018) for the convolutional weights of the encoder in order to stabilize and enhance the performance of the respective models, and in this case the models are trained for 300 epochs.

## B    RECONSTRUCTION OF NEGATIVES

Table 4: Comparing the discriminative power of VAE models with negative sampling with Bernoulli noise model, with Fashion-MNIST and MNIST as inlier and OOD datasets, respectively, when reconstruction of negative samples from EMNIST-Letters is also taken into account, with $\alpha$ weight. One can observe that the models are able to reconstruct OOD samples, while the discriminative power does not diminish. The same was observed when using Gaussian noise model.

| $\alpha$ | 0.0 | 0.1 | 0.2 | 0.3 | 0.4 | 0.5 | 0.6 | 0.7 | 0.8 | 0.9 | 1.0 |
|---|---|---|---|---|---|---|---|---|---|---|---|
| AUC BPD | 0.999 | 0.999 | 0.998 | 0.998 | 0.998 | 0.998 | 0.998 | 0.997 | 0.997 | 0.998 | 0.99 (0.01) |
| Test BPD | 0.305 | 0.304 | 0.305 | 0.306 | 0.307 | 0.307 | 0.308 | 0.308 | 0.308 | 0.309 | 0.31 (0.00) |
| OOD BPD | 1.521 | 0.681 | 0.636 | 0.639 | 0.641 | 0.632 | 0.638 | 0.634 | 0.638 | 0.645 | 0.59 (0.03) |
| AUC KL | 1.000 | 1.000 | 1.000 | 1.000 | 1.000 | 1.000 | 1.000 | 1.000 | 1.000 | 1.000 | 1.00 (0.00) |
| Test KL | 15.12 | 15.73 | 15.26 | 15.85 | 16.38 | 16.53 | 16.59 | 16.66 | 16.94 | 17.04 | 15.86 (0.23) |
| OOD KL | 325.1 | 325.4 | 325.5 | 340.2 | 350.4 | 343.0 | 354.3 | 350.1 | 353.6 | 363.2 | 321.9 (19.9) |

| $\alpha$ | Inlier Fashion-MNIST | OOD MNIST | Generated samples | Reconstructed train samples | Reconstructed test samples | Reconstructed OOD samples |
|---|---|---|---|---|---|---|
| 0.0 | | | | | | |
| 0.1 | | | | | | |
| 0.5 | | | | | | |
| 1.0 | | | | | | |

## C  INCREASING THE LATENT DIMENSION

Table 5: Comparing the discriminative performance of baseline VAE models with different latent dimension sizes, trained on Fashion-MNIST, and MNIST used as OOD dataset, with Bernoulli noise model. First column corresponds to our default setup. Reconstructed training samples and generated samples from the models are also provided. Our exploratory experiments indicate that simply increasing the latent dimension size does not help to overcome the problem of assigning higher likelihoods to OOD data, and even the generative performance is diminishing.

| Latent dimension | 10 | 50 | 100 | 250 | 500 |
|---|---|---|---|---|---|
| AUC BPD | 0.46 (0.05) | 0.35 | 0.35 | 0.39 | 0.34 |
| AUC KL | 0.61 (0.09) | 0.76 | 0.73 | 0.42 | 0.64 |
| Test BPD | 0.30 (0.00) | 0.31 | 0.32 | 0.33 | 0.34 |
| OOD BPD | 0.35 (0.08) | 0.27 | 0.27 | 0.29 | 0.29 |
| Test KL | 15.61 (0.55) | 16.06 | 16.14 | 18.02 | 17.65 |
| OOD KL | 31.91 (16.89) | 19.37 | 18.70 | 17.43 | 18.75 |
| Reconstruction | | | | | |
| Generated samples | | | | | |

# D  GENERATED SAMPLES

Table 6: Generated samples from models trained on grayscale and color images.

| Trained on Fashion-MNIST | Baseline VAE | | VAE with negative sampling negative: EMNIST-Letters | | VAE with negative sampling negative: adversarial | |
|---|---|---|---|---|---|---|
| | Bernoulli | Gaussian | Bernoulli | Gaussian | Bernoulli | Gaussian |
|  |  |  |  |  |  |  |

| Trained on MNIST | Baseline VAE | | VAE with negative sampling negative: EMNIST-Letters | | VAE with negative sampling negative: adversarial | |
|---|---|---|---|---|---|---|
| | Bernoulli | Gaussian | Bernoulli | Gaussian | Bernoulli | Gaussian |
|  |  |  |  |  |  |  |

| Trained on CIFAR-10 | Baseline VAE | | VAE with negative sampling negative: Ds. ImageNet | | VAE with negative sampling negative: adversarial | |
|---|---|---|---|---|---|---|
| | Bernoulli | Gaussian | Bernoulli | Gaussian | Bernoulli | Gaussian |
|  |  |  |  |  |  |  |

| Trained on SVHN | Baseline VAE | | VAE with negative sampling negative: Ds. ImageNet | | VAE with negative sampling negative: adversarial | |
|---|---|---|---|---|---|---|
| | Bernoulli | Gaussian | Bernoulli | Gaussian | Bernoulli | Gaussian |
|  |  |  |  |  |  |  |

# E    PLOTS FROM THE LATENT SPACE

Table 7: First two coordinates of the latent space of baseline VAE and VAE with negative sampling, with Bernoulli noise model, trained on Fashion-MNIST, and MNIST used as OOD dataset.

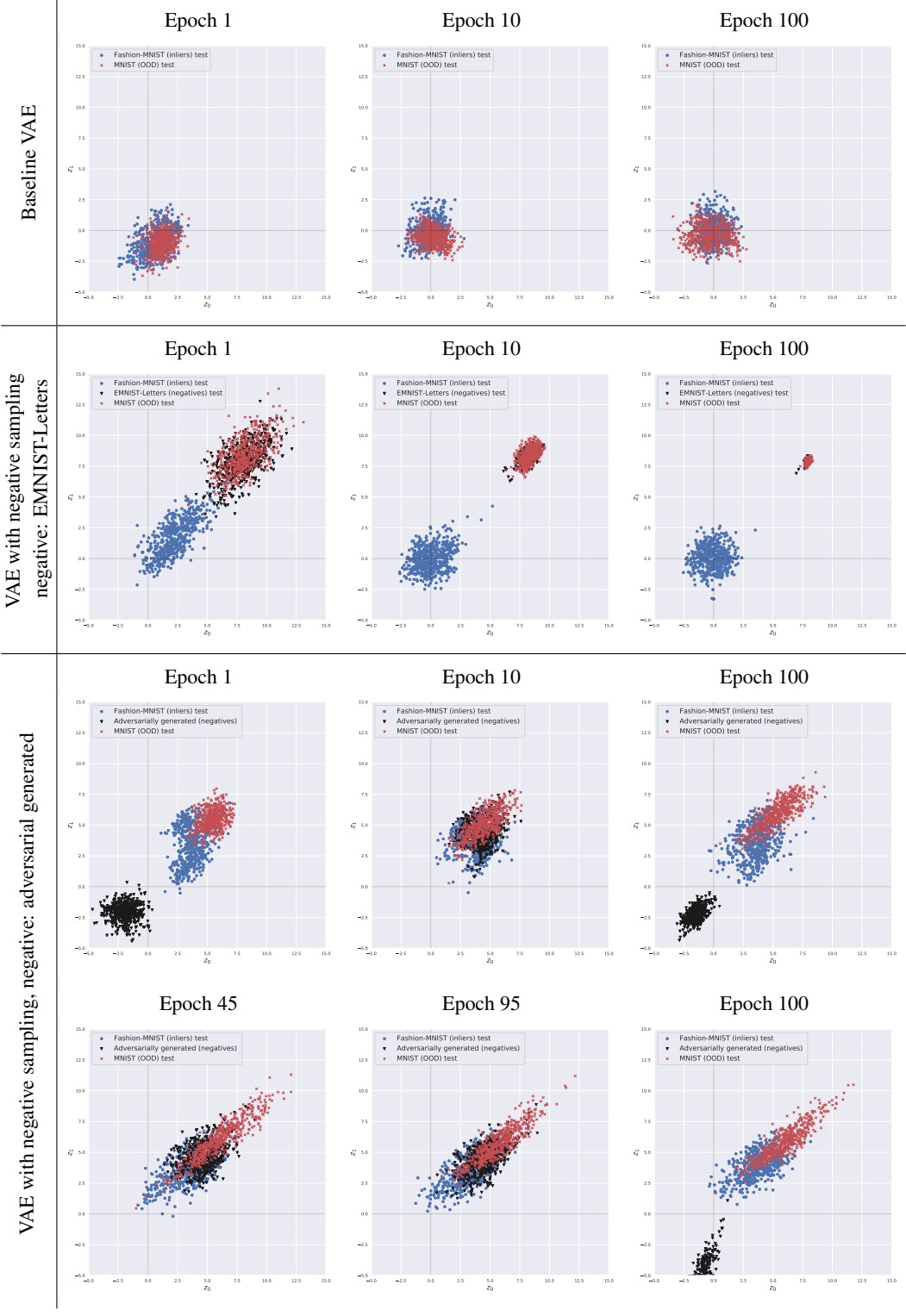

