# OpenReview forum: "Negative Sampling in Variational Autoencoders"
_ICLR.cc/2020/Conference — Reject_

### Official Review · AnonReviewer3 · 2019-10-18
**Official Blind Review #3**

**Rating:** 3

**Review:**

This paper discusses the detection of out-of-distribution (OOD) samples for variational autoencoders (VAE).
The idea is to train the encoder such that its output variational distribution q(z|\bar{x}) is pushed away from the prior of latent z.
I think the paper needs more clarification and investigation for being published in the conference.
My major concern is that more empirical investigation is necessary since the formulation provides a minor novelty.
Specific points are given below.

1) Weak novelty in terms of model design.
The objective function consists of the standard (negative) ELBO term and additional KL term to modify the variational posterior of negative samples.
This modification can be regarded as a form of outlier exposure (Hendrycks et al. 2018) specialized for VAE.
The choice of \bar{p} is not much investigated.
Any discussion if we use a more sophisticated model such as VampPrior* for stronger modeling capacity.
* J. Tomczak and M. Welling, VAE with a VampPrior, AISTATS 2018.

2) The use generated samples as negative samples is interesting but mysterious.
The authors conjecture that this works because the generated samples come from near the data manifold, but in-distribution samples and negative samples can be indistinguishable when the generative model is very well trained.
What happens if, for example, the negative samples are generated by data augmentation techniques (such as cropping, rotation, mirroring, though mirroring and much rotation may be unsuitable for text images)?
This can also produce near-manifold points.
A deeper analysis why generated samples can improve the OOD detection performance is necessary.
Furthermore, why does not this approach impact much for color images in Table 4?

3) More details of experimental procedures.
3-1) How was data points are generated from VAE as negative samples?
Possible ways are:
* sample z ~ p(z), then draw from the decoder x ~ p(x|z).
* use negative prior z ~ \bar{p}(z), then draw from the decoder x ~ p(x|z).
* this seems weird: use variational posterior z ~ q(z|x), then x ~ p(x|z).

3-2) Latent dimension of 10 for grayscale images seems small.
Does the size affect the OOD detection performance when the size is 50 or 100 to make the model richer.

3-3) How was the variance obtained when the decoder uses the Gaussian likelihood?
* fixed value?
* learned for each pixel?
* output from the decoder?

4) If we have access to diverse negative datasets, can the ODD detection perform better?
Mixing multiple datasets or using both available dataset and generated samples can improve the performance while the test OOD samples are kept unseen.
For example, train VAE on MNIST while using KMNIST and EMNIST as the negative sets to detect Fashion-MNIST as ODD.

**Experience Assessment:**

I have read many papers in this area.

**Review Assessment: Checking Correctness Of Derivations And Theory:**

I carefully checked the derivations and theory.

**Review Assessment: Checking Correctness Of Experiments:**

I assessed the sensibility of the experiments.

**Review Assessment: Thoroughness In Paper Reading:**

I read the paper at least twice and used my best judgement in assessing the paper.

---

> ### Author Response · Authors · 2019-11-15
> **Response to Review #3**
>
> We thank the reviewer for the important feedback. Based on it, we have made significant improvements on the paper. We have made clarifications and restructured the text for a better exposition and clearer message. Also, now we present even stronger experimental results and a more detailed investigation in several aspects.
>
> 1) Regarding novelty: it was not our aspiration to design an intricate model. Rather, we would like to give a simple and general approach to alleviate the bad OOD likelihood phenomenon in VAEs. We believe that our work is a valuable contribution to an ongoing discussion in the research community about out-of-distribution detection in likelihood-based models (see e.g. the works in the related work section or e.g. concurrent work submitted to this conference: https://openreview.net/forum?id=Skg7VAEKDS ).
>
> - To the best of our knowledge, we are the first to construct a training method that alleviates the bad OOD likelihoods phenomenon for VAE models. We do not just investigate and conduct detailed experiments with the Outlier Exposure technique in the VAE setting, but also present a completely new fully unsupervised approach.
>
> - We have added a short section in the paper that discusses the choice of $\bar{p}$.
>
> - Regarding more sophisticated models: please note that Nalisnick et al 2019 (https://openreview.net/pdf?id=H1xwNhCcYm ) report identical problems by other maximum likelihood models with very strong modeling capacity, such as flow-based models and PixelCNNs. Our expectation is that basically any generative maximum likelihood model is affected by these issues. It is a question of future research how best to adapt our approach to other likelihood-based models.
>
> 2) We have added a section in the paper that discusses this question, titled "why using generated data as negative samples could help?". To summarize:
>
> - Regarding the argument regarding a fully trained model, in practice, true data samples and generated samples can be distinguished even for fully trained models. But even assuming a perfect generator at convergence, during the training process the generated samples might still help to guide the model toward an equilibrium that promotes a lower likelihood for OOD samples.
>
> - Regarding data augmentation as a source of negative samples: If the augmentation actually keeps the samples within the true data manifold, then distinguishing between true and augmented data is something that we might not want to promote. The encoder would probably learn specific minor visual clues (e.g. bilinear filtering artifacts for rotations) that do not usually help assigning lower likelihood to OOD samples.
>
> - Regarding the performance of our models on color images: in the updated version of the paper we have tuned our color image models by using spectral normalization layers in the encoder. This change significantly improved the AUC values (e.g., from 0.53 to 0.85) for the models using generated negative samples, but did not improve the AUC values of the baseline models.
>
> 3-1) We employed the first option, sampling from the positive prior. We have made clarifications in the text.
>
> 3-2) Unfortunately, increasing the latent dimension does not help alleviating the bad OOD likelihood phenomenon. We have included an experiment in Appendix C that demonstrates this. We thank the reviewer for suggesting this investigation.
>
> 3-3) We have experimented with both a fixed value and a learned global value. Both cases resulted in a similar behaviour.
>
> 4) We tried to keep the experimental setup clean for the purposes of analysis, but indeed, in an engineering context we would definitely use a set of negative samples as diverse as possible.
>
> We are grateful for the valuable feedback, it greatly helped us to improve our paper.

---

### Official Review · AnonReviewer1 · 2019-10-22
**Official Blind Review #1**

**Rating:** 6

**Review:**

The paper proposes to counteract OOD problem in VAE by adding a regularization term to the ELBO. The regularizer is defined as the Kullback-Leibler divergence between a variational posterior for a negative sample and a marginal distribution over latents for negative data. The authors present experiments on MNIST and MNIST-like datasets, and CIFAR10 with SVHN. Unfortunately, I do not find the paper especially interesting. The motivation for adding the regularization term is not convincing. The experiments are insufficiently discussed.

Remarks:
- The paper proposes to ad a regularization to ELBO, namely, the Kullback-Leibler divergence between a variational distribution for a negative sample and a marginal distribution over latent variables for negative samples. I do not fully understand the motivation given on page 3. The authors show that including the negative data yields a new objective that is a sum of two log-lihelihood functions for "real" and negative data. However, later they propose to skip a (negative) reconstruction error term for the negative data. As a result, the authors obtain the objective they proposed. This explanation is very vague and I do not see what it adds to the story. Contrary, it causes new questions about their model and whether it is properly formulated.
I suggest to look into the following paper to see whether the model could be re-formulated:
Hu, Z., Yang, Z., Salakhutdinov, R., & Xing, E. P. (2017). On unifying deep generative models. arXiv preprint arXiv:1706.00550.

- I do not understand why the authors used Bernoulli distribution to model color and gray-scale images. The Bernoulli distribution could be used only for binary random variables. This is obviously flawed.

- In general, the results seem to partially confirm claims of the paper, however, they are quite vague. First, utilizing a wrong distribution is demotivating (see my previous remark). Second, I miss a better description of models and, in general, experiments' setup. Third, all results are explained in a laconic manner (e.g., "The other results in the table (...) confirm the assymetric behaviour of the phenomenon (...)"). There is neither deeper understanding nor discussion provided.

- Why there are no samples for CIFAR or SVHN provided?

======== AFTER REBUTTAL ========
I would like to thank the authors for their rebuttal. I really appreciate that the paper is updated and some concerns are solved. After reading the updated paper again, I tend to agree that the proposed idea is interesting for the problem of OOD detection using generative models. Therefore, I decide to update my score.

**Experience Assessment:**

I have published in this field for several years.

**Review Assessment: Checking Correctness Of Derivations And Theory:**

I assessed the sensibility of the derivations and theory.

**Review Assessment: Checking Correctness Of Experiments:**

I assessed the sensibility of the experiments.

**Review Assessment: Thoroughness In Paper Reading:**

I read the paper at least twice and used my best judgement in assessing the paper.

---

> ### Author Response · Authors · 2019-11-15
> **Response to Review #1**
>
> We thank the reviewer for the valuable feedback, it helped a lot to improve our paper. We have made clarifications in many places and restructured the text for a better exposition and a clearer message.
>
> In our humble opinion, our results are easier to appreciate in the fuller context of the growing amount of work related to out-of-distribution detection in likelihood-based models (the works in the related work section or e.g. concurrent work submitted to this conference: https://openreview.net/forum?id=Skg7VAEKDS ). Our contributions reflect on recent work, and provide several novelties:
> - To best of our knowledge, we are the first to give a training method for VAEs that helps alleviating the bad OOD likelihood performance.
> - We present an unsupervised approach that is completely novel, and report detailed experiments that confirm the robustness and usefulness of the method (see Table 1 and Table 2 in the updated paper).
> - Our work highlights a potential problem with utilizing Outlier Exposure (the very general framework laid down by Hendrycks at al. 2018). The results with auxiliary datasets in Table 1 show that while auxiliary samples help greatly in most cases, OOD detection performance can be very sensitive to the choice of the auxiliary dataset, see for example the last block of Table 1, Letters-Fashion-Numbers, where Outlier Exposure fails to improve while our adversarial method still achieves good performance.
>
> - We have expanded the description and the discussion of the experiments which now also continues in the appendix.
>
> - We have updated the explanations and motivations in several places where the review identified issues.
>
> - We have added an experiment in the appendix which explores the utilization of the reconstruction term for the negatives during training. Our experiments show that this does not improve the model in terms of OOD performance.
>
> - Regarding Bernoulli: we agree with the reviewer that the Bernoulli is a theoretically less sound modeling choice than, for example, the Gaussian. (We added a footnote to the paper with this remark.) That's one of the reasons we publish numbers with a Gaussian noise model as well. However, much of the literature directly relevant to us made the exact same modeling choice: working with the Bernoulli, and interpreting the grayscale values as probabilities of binary events. Loaiza-Ganem and Cunningham 2019 https://arxiv.org/abs/1907.06845 lists several papers following this practice. We side with the reviewer in this disagreement. However, as we said above, the Bernoulli is an unavoidable option if we wish to compare our results to the rest of this sub-field.
>
> - We thank the reviewer for the pointer for the Hu et al. paper. As we see, the main contribution of our paper is not in proposing a hybrid VAE-GAN model, but in tackling the issue of bad OOD likelihoods both in a supervised and unsupervised context. We clarified in the text how the adversarial model is trained.
>
> - We have added sample images for all datasets to Appendix D.
>
> We are very thankful for the review, it helped us a lot to improve the paper.

---

### Official Review · AnonReviewer2 · 2019-10-23
**Official Blind Review #2**

**Rating:** 3

**Review:**

Summary:
The authors propose augmenting VAEs with an additional latent variable to allow them to detect out-of-distribution (OOD) data. They propose several measures based on this model to distinguish between inliers and outliers, and evaluate the model empirically, finding it successful.

Unfortunately, the method in this paper is developed unclearly and incorrectly. Although their experiments are somewhat successful, the problems with the text and method are severe enough to justify rejection.

Specifically, the authors' method proposes adding a term to the loss of the VAE that encourages the variational posterior (q) to distribute latent codes (z) for inliers and outliers differently. The equation which defines their new objective is unclear -- specifically, it is not clear whether the added KL term is computed for inliers and outliers both, or whether it is only computed for outliers. If it is the former, then the method does not make sense. If it is the latter, then the equation is incorrect or at the very least not clear in the extreme.

Furthermore, the term is added without consideration of whether or not the method is still optimizing a sensible variational lower bound. The authors attempt to justify the objective by writing out a variational lower bound for a VAE with a mixture prior where inliers and outliers are generated from different mixture components. However, their equations are incorrect -- the equation that is called the log likelihood is not the log likelihood, and the ELBO is similarly wrong.

Their empirical evaluation is reasonable, although the measures they propose to distinguish between inliers and outliers (i.e. the kl from the approximate posterior to the prior) is not thoroughly justified.


**Experience Assessment:**

I have published in this field for several years.

**Review Assessment: Checking Correctness Of Derivations And Theory:**

I carefully checked the derivations and theory.

**Review Assessment: Checking Correctness Of Experiments:**

I assessed the sensibility of the experiments.

**Review Assessment: Thoroughness In Paper Reading:**

I read the paper thoroughly.

---

> ### Author Response · Authors · 2019-11-15
> **Response to Review #2**
>
> We are grateful to the reviewer for the insightful comments. We have rewritten large parts of the text with the goal of making the descriptions of our models and our core claims more clear. Just as importantly, we have an updated experiments section with even stronger results. (See e.g. Table 1 or Table 2 in the updated paper, which highlight the effectiveness of our proposed method.)
>
> - Regarding the KL term: indeed, the latter interpretation is the intended one. $\bar{x}^{(i)}$ is specified to be a negative sample, and the extra term only references $\bar{x}^{(i)}$, not $x^{(i)}$. We have made clarifications in the text.
>
> - We are deeply thankful to the reviewer for pointing out that we made a mistake in writing up a variational model justifying our loss function. To our defense, this toy model was not central to our argument. We have removed it from the paper.
>
> - Regarding the positive KL term as a measure: examining the VAE likelihood estimates raises the question of how the two components of the ELBO (the reconstruction part and the KL part) contribute to the likelihood estimate and the discriminative power of the model. As the magnitude and the behavior of the reconstruction term are highly determined by the choice of the noise model, it is natural to investigate to what extent the separation between inliers and outliers is carried out in latent space. Note that we publish likelihood-based evaluations everywhere, one can consider the KL-based evaluations as extra information.

---

### Author Response · Authors · 2019-11-15
**Paper update**

We thank the reviewers for their many helpful comments. Incorporating them improved the paper tremendously, and we apologize in advance for pushing the limits of how much a paper can change during the rebuttal period.

We have uploaded a new version of the paper with significant improvements:
- we further strengthen our experimental results: our new measurements show that our models improve on the baselines in a very consistent manner,
- we have restructured the text for a clearer exposition and presentation,
- we have removed the erroneous claim from Section 3. We thank AnonReviewer2 for pointing it out.

Based on our detailed answers and the results of the new version, we kindly ask the reviewers to reassess their evaluation.

---

### Comment · Area_Chair1 · 2019-11-15
**Reviewers, any comments on the author response?**

Dear Reviewers, thanks for your thoughtful input on this submission!  The authors have now responded to your comments.  Please be sure to go through their replies and revisions.  If you have additional feedback or questions, it would be great to know.  The authors still have one more day to respond/revise further.  Thanks!

---

### Decision · Program_Chairs · 2019-12-19

**Decision:**

Reject

**Comment:**

This paper proposes to improve VAEs' modeling of out-of-distribution examples, by pushing the latent representations of negative examples away from the prior.  The general idea seems interesting, at least to some of the reviewers and to me.  However, the paper seems premature, even after revision, as it leaves unclear some of the justification and analysis of the approach, especially in the fully unsupervised case.  I think that with some more work it could be a very compelling contribution to a future conference.